# Supporting health and social care professionals in serious illness conversations: Development, validation, and preliminary evaluation of an educational booklet

Silvia Gonella[1]*, Paola Di Giulio[2], Federica Riva-Rovedda[2]*, Luigi Stella[3], Maria Marcella Rivolta[4], Eugenia Malinverni[5], Mario Paleologo[1], Giancarlo Di Vella[2], Valerio Dimonte[1,2]

1 City of Health and Science University Hospital Turin, Turin, Italy, 2 Department of Public Health and Pediatrics, University of Torino, Turin, Italy, 3 Fondazione Assistenza e Ricerca Oncologica (F.A.R.O.), Turin, Italy, 4 Local Health Authority of the City of Turin, Turin, Italy, 5 Fondazione Luce per la Vita Onlus, Rivoli, Turin, Italy

* silvia.gonella@unito.it (SG); federica.rivarovedda@unito.it (FRR)

**Data Availability Statement:** All relevant data are within the paper and its Supporting Information files

## Abstract

Serious illness conversations aim to align the care process with the goals and preferences of adult patients suffering from any advanced disease. They represent a challenge for healthcare professionals and require specific skills. Conversation guides consistent with task-centered instructional strategies may be particularly helpful to improve the quality of communication. This study aims to develop, validate, and preliminarily evaluate an educational booklet to support Italian social and healthcare professionals in serious illness conversations. A three-step approach, including development, validation, and evaluation, was followed. A co-creation process with meaningful stakeholders led to the development of the booklet, validated by 15 experts on clarity, completeness, coherence, and relevance. It underwent testing on readability (Gulpease index, 0 = lowest-100 = maximum) and design (Baker Able Leaflet Design criteria, 0 = worst to 32 = best). Twenty-two professionals with different scope of practice and care settings evaluated acceptability (acceptable if score ≥30), usefulness, feasibility to use (1 = not at all to 10 = extremely), and perceived acquired knowledge (1 = not at all to 5 = extremely). After four rounds of adjustments, the booklet scored 97% for relevance, 60 for readability, and 25/32 for design. In all, 18 (81.8%), 19 (86.4%) and 17 (77.3%) professionals deemed the booklet acceptable, moderate to highly useful, and feasible to use, respectively; 18/22 perceived gain in knowledge and all would recommend it to colleagues. The booklet has good readability, excellent design, high content validity, and a high degree of perceived usefulness and acquired knowledge. The booklet is tailored to users' priorities, mirrors their most frequent daily practice challenges, and offers 1-minute, 2-minute and 5-minute solutions for each scenario. The co-creation process ensured the development of an educational resource that could be useful regardless of the scope of practice and the care setting to support professionals in serious illness conversations.

**Funding:** This work was supported by Fondazione Assistenza e Ricerca Oncologica (F.A.R.O.) Onlus. The funding did not influence the study process either the study findings. The funder had a role in the data collection, but not in study design, data analysis, decision to publish or preparation of the manuscript.

**Competing interests:** The authors have declared that no competing interests exist.

## Introduction

Serious illness conversations represent a wide range of discussions engaging healthcare professionals, patients and/or their informal caregivers when a health condition with a high risk of mortality and a negative impact on quality of life occurs or excessively strains the caregivers [1]. Serious illness conversations need a complex skillset that has been conceptualized as both a skill and as a way of being in relation to the other [2]. Indeed, such conversations occur in a context of uncertainty and turbulent emotions, and involve communication skills as well as relational skills of empathy, flexibility, and self-reflection [3]. According to their goal, serious illness conversations require competencies in delivering news, eliciting hopes and worries, sharing or clarifying the prognosis, promoting illness understanding, exploring what is important for the patient and their goals and wishes for care, making recommendations, but also in responding to emotions and offering support and reassurance [4, 5].

Although there is consensus that effective communication plays a key role in the establishment of therapeutic alliance and partnership based on mutual trust [6] and contributes to know the patients' valued priorities and align care with their preferences [7], communication skills training is generally not part of traditional education for healthcare professionals in Italy, especially for clinicians. As a consequence, the novices who enter the profession often feel underprepared and unequipped for difficult conversations [8], and tend to postpone them as much as possible [9]. However, there is increasing awareness that communication skills are both innate and teachable [6, 10] and communication skills training is a growing component of formal professional training for healthcare professionals in other healthcare systems such as the UK, US, Canada and Australia, where programs to promote high quality communication have been introduced in the education of healthcare students and also represent a consolidated component of continuing education for professionals [11–14].

Several institutions are recognizing the worthy contribution of effective communication in avoiding aggressive, non-beneficial treatments and delivering high-quality care [15], with increasing investments in communication skills training [16, 17]. At least at a basic level, all members of the team should be confident and capable of engaging in serious illness conversations [18].

Experts in the field of communication skills recognize that didactic material can support experiential learning [10]. Printed educational resources are among the most common approaches to translate research into clinical practice tips [19]. They may facilitate clearer communication by providing a mental model of common clinical scenarios (i.e., context, goals, challenges to overcome, strategies to be used, etc.) [6, 18]. A written conversation guide offers structure for the conversation and facilitates timely, effective conversations [17, 20], while maintaining an active role of professionals [4]. Compared to other formats such as video or audio guides, written guides are more quickly accessible in clinical contexts, make it easier to revisit a particular section, and their computerized version allows one to search for keywords that make them more discoverable [19]. When conversation guides were used, professionals perceived that they had a framework for difficult conversations [17] and their satisfaction with their role increased [20].

Written conversation guides have been introduced in different clinical contexts, such as emergency departments, intensive care units, primary care, oncology, and long-term care to provide high-quality patient-centered care [12, 21, 22]. In these experiences, the original conversation guides were adapted to the care environment and the scope of practice of the healthcare professionals who would use the guide [23].

Conversation guides that are consistent with task-centered instructional strategies may be particularly helpful to improve the flow and speed of conversation [17]. The education theory of task-centered instructional strategies assumes that optimal learning occurs with focused

tasks that learners are likely to encounter in their daily practice, associated with readily accessible procedural information [24]. The former makes it clear what the learner should do, while the latter provides the learner with the information they need to perform the task and decreases their cognitive load [24].

Although it is acknowledged that there is no "magic bullet" to change professional practice [25] and that the context influences the effectiveness of strategies [26], twelve key theoretical constructs have been identified to explain healthcare professionals' behavior change [27]. In order to promote behavior change, interventions should enhance (1) knowledge and (2) skills, (3) be consistent with social/professional role and identity, (4) serve as persuasive strategies which influence beliefs about capabilities and (5) lead to changes in beliefs about consequences, (6) stimulate reflection on motivation and goals, (7) influence memory, attention, and decision processes, (8) create a facilitative environmental context that triggers behaviors, evoke (9) emotions and (10) behaviours while providing regulation, and (11) propose desirable behaviours and (12) social norms.

In the effort to maximize the benefit of an innovative communication skills training program for different health and social care professionals [28], as an integral part of the intervention, we developed an educational booklet aimed at providing professionals with guidance in challenging communication scenarios with frail or terminally-ill patients and/or their families. Unlike previous conversation guides [29, 30] that provide a talking map with a series of signposts to cover the conversation flow with no reference to clinical cases, this booklet is built on complex communication scenarios (e.g., unawareness of disease trajectory or prognosis, ambivalent care preferences) based on realistic cases. This is likely to improve learners' memory of content because they immediately perceive support for real-case scenarios. Moreover, previous guides employ only one communication strategy; instead, having more solutions could maximize the chance for learners to find the communication strategy that best fits their learning style (e.g., short tips, infographics, or illustrative statements). Therefore, the booklet adopts several communication strategies and covers the most common and challenging communication scenarios that professionals can experience in clinical practice to help them acquire knowledge of their real-word tasks and form mental models for how integrating the skills into their performance. In fact, relevance is a pivotal component of motivation for behavioural change [31].

The primary aim of this study was to describe the development and validation of the educational booklet. The secondary aim was to assess its acceptability, usefulness, and feasibility to use in daily practice and the acquired knowledge on communication techniques perceived by professionals.

## Material and methods

This study was carried out according to a three-step approach [32] and adhered to the guidelines outlined in the Standards for QUality Improvement Reporting Excellence in EDUcation (SQUIRE-EDU) [33] (S1 Table).

The methodology included the creation of the resource, validation, and evaluation. The first step entailed a literature review, qualitative interviews, and a co-creation process to develop the booklet. The second step involved the booklet's validation (i.e., relevance) through health and social care professionals; moreover, completeness, clarity, realism of clinical scenarios, readability, and design were assessed. The third step concerned the preliminary evaluation of the booklet by health and social care professionals working in different clinical settings. The professionals rated the acceptability, usefulness, and feasibility of using the booklet in daily practice as well as their acquired knowledge on communication techniques and protocols after

reading the booklet. These steps ensured the methodological validity and reproducibility of this study (Fig 1).

Participants involved in the development of the booklet provided verbal consent that was audio and video recorded, while those involved in the validation and preliminarily evaluation steps provided written consent. The study was approved by the Ethics Committee of the University of Torino (n. 0598416/2021).

## First step: Development of the booklet

A first draft of the booklet covering the most common and difficult communication scenarios was created according to a review of the literature [8, 34] and interviews with health and social care professionals with varying scopes of practice (*findings published elsewhere*) [35]. The first draft covered nine difficult communication scenarios against four clinical cases in different contexts of care where difficult conversations are frequent: 1. Carlo, 77 years, with advanced dementia in nursing home; 2. Giovanni, 80 years, with lung cancer in intensive care unit; 3. Ottavio, 47 years, with acute lymphoblastic leukemia in hematology; 4. Anna, 85 years, transferred to long-term services after in-hospital stay for heart failure. Communication scenarios

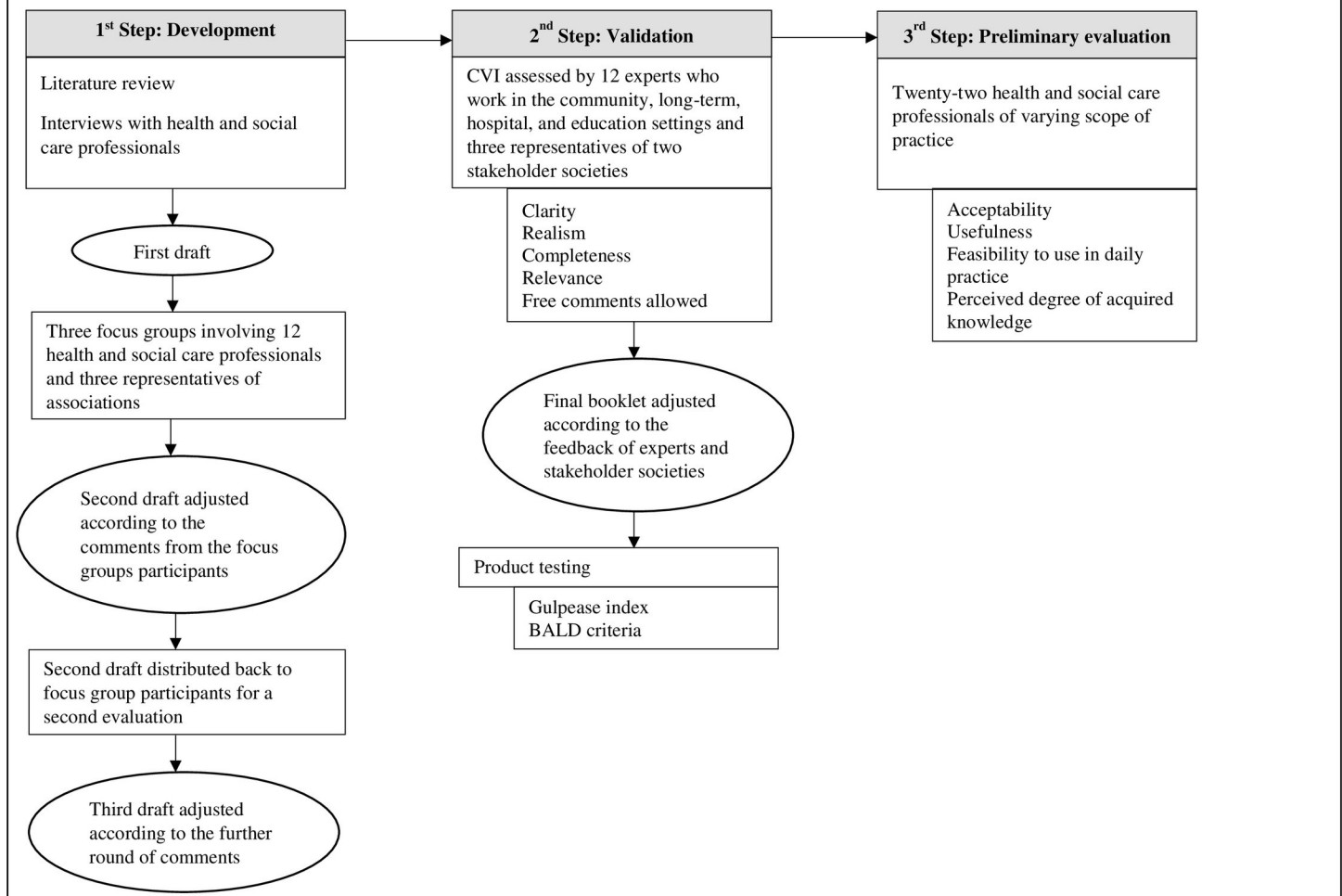

**Fig 1. Development, validation, and preliminary evaluation of the educational booklet: A three-step approach.** BALD, Baker Able Leaflet Design; CVI, Content Validity Index.

included unawareness of disease trajectory or prognosis, turbulent emotions and prolonged silences, preferences for aggressive care, ambivalent care preferences, family carers with different care preferences, choosing care treatments when the person's preferences are unknown, unrealistic expectations, balancing hope and realism, and remote communication. Each scenario is accompanied by i) practical hints to deal with serious illness conversations; ii) infographics summarizing the main evidence-based indications; and iii) good examples of communication scripts based on the most known communication protocols (i.e., CONNECT, SPIKES) and methods (i.e., ADAPT, NURSE, PREPARED, REMAP, TUVERI). All these protocols and methods share core components such as paying attention to the context, exploring patient/family perceptions and preferences and providing information tailored on their needs, and addressing emotions with empathic responses. Each protocol/method also has a specific focus: ADAPT and PREPARED were designed to provide a talking map in discussing prognosis; the NURSE method illustrates communicative strategies for responding to emotions; the REMAP tool may be useful to address goals of care; the CONNECT protocol was designed to organize key aspects of remote communication; finally, SPIKES and TUVERI provide guidance in disclosing unfavorable information. A detailed review of the principles or content of the main communication protocols and methods is shown in S2 Table.

An introductory text, a section on basic communication skills, and additional resources complete the booklet.

Secondly, 12 health and social care professionals and three representatives of patients associations, family carers, and volunteers in the field of dementia or palliative care were recruited between 5 September 2022 to 26 October 2022 and participated in three focus group discussions (5 participants for each focus group). The research team reached the reference person of each association that then identified their representative. Characteristics of the participants to focus groups are shown in S3 Table.

Seven to ten days prior to the focus group, each participant received the booklet by email and was asked to read it for sharing ideas on the content during the group discussion. The focus groups were conducted digitally to facilitate participation from different geographical areas [36] and had a mean duration of 78 minutes (range 68–85). A member of the research team moderated the discussion using a pre-defined set of questions and solved conflicts of opinion by seeking common ground and areas of agreement to compromise. The questions guiding the focus groups explored the completeness, acceptability, clarity, feasibility to use, and graphics of the booklet, that the literature recognizes as key features for written resources [19]. Another team member took any field notes on paraverbal and body language. Focus group discussions were audio-recorded, transcribed verbatim and transcripts checked for accuracy. Then, two researchers used an inductive thematic analysis to analyse the transcripts and identify the areas of improvement [37]. Thirdly, this analysis guided the changes of the second iteration of the booklet, which was then distributed back for a second evaluation with the request to complete a brief open-ended questionnaire assessing its content and design. Fourthly, the booklet was further adjusted according to the emerging comments.

The research team initially took the decision not to involve a design agency and to deal on their own with design issues that could have arisen. However, the team later realized that it was not possible to address satisfactorily criticalities about font, colors, and layout as they recurred on the unfolding versions of the booklet despite the efforts to solve the problems; therefore, a graphic designer was involved after the feedback from the external review board.

## Second step: Validation

**Content validity index.** An external review board of 12 health and social care professionals who work in the community, long-term, hospital, and education settings and three representatives of two stakeholder societies reviewed the booklet and answered an online, 10-minute questionnaire from 17 January 2023 to 17 February 2023. Characteristics of the board are shown in the S4 Table.

The professionals received the booklet by email and were invited to rate each section according to its clarity, coherence, and relevance. Moreover, the booklet was explored for completeness and missing information. Clarity, coherence, and relevance of each section were assessed using a 4-point Likert scale (from 1 = not at all to 4 = much). The completeness was based on a dichotomous answer (yes/no) with the opportunity to list missing information. Free comments were also invited.

Regarding the relevance of the booklet, the content validity index (CVI) was computed for individual sections (I-CVI) and for the overall booklet (S-CVI/averaging calculation method (Ave) and S-CVI/Universal Agreement (UA)) [38].

The I-CVI was calculated for each section as the number of professionals giving a rating of either 3 or 4 (thus relevant), divided by the total number of professionals. With more than 10 validators, the I-CVI should be >0.70 [38]. Only sections scoring higher than this threshold were maintained [38].

The S-CVI/Ave was computed as the average of the I-CVIs for all sections on the booklet. A S-CVI/Ave of 0.90 or higher is recommended. The S-CVI/UA was computed as the proportion of sections of the booklet that achieved a relevance rating of 3 or 4 by all the validators. A S-CVI/UA of 0.80 or higher is recommended [38].

The clarity and coherence of each section and for the overall booklet were computed as mean with standard deviation (SD). The completeness of the booklet was computed as the number of validators who rated the booklet as complete on the total.

When a section was marked as 'not at all' = 1 or 'little' = 2 for relevance, clarity, or realism of clinical scenarios by at least two validators, the text was adjusted according to their comments.

**Product testing.** The readability, layout, and design of the final booklet were evaluated. The Gulpease Index was used to assess the readability [39]. The Gulpease index considers two linguistic variables, the length of the word and the number of letters of the sentence, as shown in the following formula: $89 + \frac{300*(number\ of\ sentences) - 10*(number\ of\ letters)}{number\ of\ words}$ [39]. Accounting for the length of a word in characters rather than in syllables, it is proved to be more reliable in assessing the readability of Italian texts. The index ranges from 0 (lowest readability) to 100 (maximum readability). Scores below 80, 60 and 40 identify texts that are difficult to read for a 5th grade level (primary school level, 6 to 10 years), 8th grade level (junior secondary school level, 11 to 13 years), and 13th grade level (secondary school level, 14 to 18 years) [40]. Each section of the booklet was tested at the following address: https://farfalla-project.org/readability_static/.

The Baker Able Leaflet Design (BALD) criteria were used to assess the layout and design characteristics of the booklet [41]. The BALD score covers 16 attributes and ranges from 0 (worst layout and design) to 32 (best layout and design).

## Third step: Preliminary evaluation

**Recruitment of health and social care professionals.** Health and social care professionals were recruited at a 800-bed acute care hospital and its community services and at two nursing

homes in the North-west of Italy from 10 May 2023 to 15 May 2023. Professionals of any profile engaged in serious illness conversations with patients or their families were eligible.

**Questionnaire.** The professionals were asked to complete a questionnaire that explored their perception on the acceptability, usefulness, and feasibility of using the booklet in daily practice. Moreover, they were asked to rate their acquired knowledge on communication techniques and protocols and could suggest changes. Finally, professionals were asked whether they would recommend the booklet to colleagues (yes/not) and to motivate it.

Acceptability is referred to by O'Connor and Cranney as "ratings regarding the comprehensibility of components of a decision aid, its length, amount of information, balance in presentation of information about options, and overall suitability for decision making" [42]. Our measure of 'acceptability' refers more specifically to acceptability to use, understand, read, and inform patients and their families. It also includes the themes of communication with patients/families, relationships with patients/families, and intention to use the booklet in practice. For this, we developed a 10-item scale summing rating of agreement with statements selected from the acceptability instrument by O'Connor and Cranney [42] and from a comparative evaluation of three decision aids [43]. Item scores were 1 (strongly disagree) to 5 (strongly agree) [42]. Higher total scores represent better acceptability. We regarded scores of 30 and higher as "acceptable" and 40 and higher as "highly acceptable" (mean item scores 3 and 4, respectively).

Usefulness and feasibility to use were rated on a scale from 1 (not at all useful/feasible to use) to 10 (extremely useful/feasible to use). The mean usefulness and feasibility scores were calculated; a score between 6 and 7 indicated moderate usefulness/feasibility, while a score >8 a high degree of usefulness/feasibility. Acquired knowledge was rated on a 5-point Likert scale from 1 (not at all) to 5 (extremely).

## Statistical analyses

Frequencies, percentages, and means with range or standard deviation (SD) were computed. IBM SPSS Statistics for Window Version 27.0 was employed. The sample size for the evaluation step was based on the concept of content validity and was calculated using the formula: $n = \frac{z^2 P(1-P)}{e^2}$, where 'P' is the expected proportion of health and social care professionals who indicated the relevance of each section of the booklet and 'e' represents the acceptable proportional difference compared to what would be expected [44]. An expected proportion of professionals of 85% was established with a difference of 15% and a confidence level of 95%. This results in a minimum of 22 professionals being needed [44]. This analysis is based on the methodology for the evaluation of decision aids. Decision aids and educational booklets share several similarities, including the multidisciplinary approach, evidence-based soundness, and attention to both content and design. However, some differences also occur: decision aids explicitly state the decision that needs to be considered and appear as active tools to guide the decision-making process by providing detailed focus on options and outcomes and helping people to arrive at decisions that reflect their personal values and preferences. Instead, educational booklets represent a source of information but are not focused on decision points due to their broader perspective [45].

## Results

### First step: Development of the booklet

The inductive thematic analysis of focus group transcripts identified eight areas for improvement: 1) Make clearer that the booklet is part of a communication skills training program; 2)

Strengthen the relational component of communication; 3) Enhance the realism of clinical scenarios; 4) Draw attention to the legal issues; 5) Insert missing critical situations; 6) Employ a person-centered dialogical approach; 7) Enhance actionability; and 8) Improve graphics.

Further refinements came after re-evaluation by focus group participants and external reviewers, but without major changes.

Table 1 shows the main adjustments of the booklet during its development.

The final version of the booklet is freely available online and is downloadable from the website of the Italian Society of Palliative Care at https://www.sicp.it/aggiornamento/linee-guida-bp-procedures/2023/09/guida-alla-comunicazione-in-situazioni-complesse/

## Second step: Validation

**Content validity index.** All external reviewers deemed the booklet complete with an overall mean (SD) clarity and coherence of 3.7 (0.5) and 3.9 (0.3), respectively. The final booklet resulted into 17 sub-sections and none was removed based on I-CVI always above 0.9. No differences emerged between professional raters and association representatives. The S-CVI/Ave was 0.97 and the S-CVI/UA was 0.53. The S5 Table shows the details for each sub-section.

**Product testing.** The mean Gulpease index of all sub-sections was 60. This represents a language fairly easy to read and understand at a junior secondary school level. The 'Introduction' sub-section scored the lowest with 44. The 'Actionable resources' subsection scored the highest with 71, followed by 'Basic communication skills' with 70 (S5 Table).

The BALD score for the final version of the booklet was 25/32. The S6 Table shows the detailed design characteristics of the booklet.

## Third step: Preliminary evaluation

**Participants.** Twenty-two professionals were contacted and all answered the evaluation questionnaire. The majority were females and the over 35 to under 35 years ratio was roughly equal. More than two thirds (n = 15) had more than 10 years of working experience with only three professionals with less than one year in the current service. Most were nurses, however, the profiles of physician, social worker, occupational therapist, and physiotherapist were also represented. The professionals worked in varying settings of care, including emergency, nursing home, medicine, surgery, primary care, and hospice. Only three professionals undertook specific education in clinical communication or counselling (Table 2).

**Questionnaire.** The evaluations for acceptability, usefulness, feasibility to use, and perceived acquired knowledge are presented in Table 3. In all, 18 (81.8%), 19 (86.4%) and 17 (77.3%) professionals deemed the booklet acceptable, moderate to highly useful, and moderate to highly feasible to use, respectively. Over 80% (n = 18) of the professionals perceived gain in knowledge after reading the booklet.

Table 4 shows the acceptability score at the item level: particularly, professionals perceived the booklet as consistent with their professional identity and standard by agreeing that it aligns with the way they think things should be done (15/22) and complements their usual approach to communication (14/22). All but three professionals perceived a high probability that using this booklet may result in more benefit than harm, and half believed that the booklet would improve communication and relationships between professionals and families. Around 60% (13/22) reported that they would continue to use the resource after the end of the project.

All professionals would recommend the booklet to colleagues and 11 reported reasons: "provides practical and easily applicable advice" (n = 6), "provides a mental model and guidance in real-case scenarios" (n = 3), "helps start conversations" (n = 1), and "promotes

**Table 1. Adjustments of the booklet during the process of development.**

| Sections of the booklet | First draft | Second draft | Third draft | Final version |
|---|---|---|---|---|
| Title | Communicating at the end of life: A practical guide for healthcare professionals dealing with difficult communications | Communicating in complex situations: A practical guide for health and social care professionals | No additional changes. | No additional changes. |
| Introduction | Short introduction (226 words) that describes the targeted audience and aim of the booklet, benefits of effective communication and feasibility to learn communication skills and acquire communication competence. | Longer introduction (905 words) that better frames the booklet as a supportive tool promoting awareness, empowerment, accountability, and reflection on communication in complex situations. The relational aspect of communication has been valorized. The role of experience is recognized as well as the impossibility to squeeze communication in strict protocols that rather provide mental models to tailor the interaction according to the specific situation. | It has been further clarified that this booklet does not cover the complex and multiple nuances of the reality but rather it is a starting point for effective communication where experience and interpersonal skill are put at stake. | Redundancies across communication protocols have been noted. This led to clarify that protocols are largely overlapping for the main core contents and the choice of which protocol to adopt depends on personal preferences. |
| Basic communication skills | The main basic communication skills and techniques (e.g., open questions, paraphrasing, summarizing, "tell me more about, "I wish") are presented with examples. | Didactic and textbook examples have been replaced with expressions drawn from the real experience of the professionals. | No further changes. | This section was judged too technical with little attention to the relational aspect. Therefore, the need to prepare the setting of communication as well as the pivotal role of team collaboration and the need for professionals to be physically, emotionally, and psychologically mindful has been emphasized. |
| Difficult communication scenarios | Nine scenarios against four clinical cases (Carlo, Ottavio, Giovanni, and Anna) in different settings of care. Each scenario is accompanied by i) good examples of communication scripts based on the most popular communication protocols (i.e., CONNECT, PREPARED, REMAP, SPIKES) and methods (i.e., ADAPT, NURSE, TUVERI); ii) practical hints to deal with serious illness conversations; and iii) an infographic summarizing the main evidence-based indications. | Each communication scenario now opens with an evocative sentence to convey the message that reality is complex. The realism of dialogues has been improved replacing didactic and textbook sentences with expressions drawn from the real experience of the professionals. For example, to explore family carers' perspective about the clinical conditions of their beloved, the original sentence *"Could you explain better what you mean when you say you didn't expect it?"* was changed into *"I understand that you were not prepared for this worsening of your husband. Is there anything you would like to ask me?"*. To expect emotion and empathize, the original sentence *"I can imagine how you feel and how it's difficult having to make a decision like this. What worries you? Would you like to talk about it?"* was changed into *"I realize this news has shocked you and it's difficult having to make a decision like this. My colleagues and I are available if you would like to talk about it"*. To promote reflection on care choices, the original sentence *"We have given you a lot of information. If you agrees, we would continue the discussion tomorrow also with your mother and your sister"* was changed into *"You have received a lot of information. It would be important to come together with a decision that guarantees the best interest of your father and what he would have wanted without involving the judge supervises cases concerning guardianship"*. | Some sentences regarding the role of patients and their family carers in taking care decisions have been reworded and the necessity to involve the patient in the decision-making process, if possible, has been highlighted. | The order of communication scenarios in Carlo's case - unawareness of disease trajectory or prognosis, turbulent emotions and prolonged silences, preferences for aggressive care, family carers with different care preferences, choosing care treatments when the person's preferences are unknown- was modified by shifting "choosing care treatments when the person's preferences are unknown" between "turbulent emotions and prolonged silences" and "preferences for aggressive care" to reflect a temporal logic. Also, in the scenario "family carers with different care preferences", it has been clarified the role of family carers in care decisions when the relative is no more cognitively competent according to the Italian law by adding the following sentence "This is a decision that awaits us healthcare professionals, you don't have the responsibility for the choice. We ask you to help us understand what your father would have wanted". Each check-list has been associated with a QR code which redirects to the desired information when smartphones and tablets frame the code. |

*(Continued)*

**Table 1.** (Continued)

| Sections of the booklet | First draft | Second draft | Third draft | Final version |
|---|---|---|---|---|
| | | To summarize the key points discussed, the original sentence *"Could you try repeating in your own words what we told you?"* was changed into *"Before closing the encounter, if you agree, we would like to try to summarize the main points we discussed…"*.<br>An extensive rewording consistent with a person-centered approach has been performed to highlight that communication is a mutual dialogue and professionals have to bring out preferences, welcome, support and establish trusting relationships.<br>Legal issues have been emphasized by clarifying the role of the patient, their family carers, and professionals in the decision-making process, as well as the need to document the process of communication beyond its outcomes.<br>A sub-section listing other possible complex communication scenarios has been added to further stress that the booklet is focused on the most common ones. | | |
| Additional resources | This section consists of four sub-sections: i) Running the family conferences; ii) Summary of the infographics for each communication scenario; iii) Flow charts summarizing the process of communication in presence and at distance; and iv) Communication protocols and communication methods. | A person-centered approach has been adopted in the flow charts by reporting first extracts of dialogues from communication scenarios of the previous section and then referring to the communication protocol(s). Also, the concept of "documenting the process of communication" has been introduced in the flow charts.<br>A further sub-section on times and manners of communication has been added. | No further changes. | No further changes. |
| Actionable resources | Absent | The summary of the infographics has been moved from the "Additional resources" section to a newly-added "Actionable resources" section. The check-lists appear as tear-away, ready-to-use worksheets. | No further changes. | No further changes. |
| Layout and design | Different fonts and font size (range 10 to 14) for check lists, communication scripts, and narrative text.<br>Abstract or stylized images.<br>White background with leaves-based pattern. | Increased font size across the entire booklet.<br>Abstract and stylized images substituted with pictures highlighting the importance of human relationships.<br>Light yellow background with removal of the leaves-based pattern.<br>Page numbers added. | Further increase in the font size and change of two images. | Three members of the external review board highlighted some criticalities about font, colors, and layout therefore a graphic designer was involved to develop a better and user-friendly version of the booklet. |

reflection on professional competence and learning needs" (n = 1). Only one professional suggested enriching the bibliography to improve scientific soundness.

## Discussion

This study describes the development, validation, and preliminarily evaluation of an educational booklet aimed at supporting Italian professionals in serious illness conversations. The co-creation process led to an acceptable resource with valid content, high degree of perceived usefulness, feasibility to use, and acquired knowledge.

The co-creation process for developing the booklet relied on focus groups involving a purposive sample of stakeholders. Focus groups represent a person-centered research method that

**Table 2. Characteristics of professionals evaluating the educational booklet (n = 22).**

| Participants characteristics (n = 22) | N |
|---|---|
| **Female gender** | 18 |
| **Age,** years | |
| ≤ 25 | 2 |
| 26–35 | 8 |
| 36–50 | 9 |
| > 50 | 3 |
| **Overall working experience,** years | |
| 1–4 | 3 |
| 5–10 | 4 |
| 11–15 | 5 |
| > 15 | 10 |
| **Experience in the current service,** year | |
| < 1 | 3 |
| 1–4 | 9 |
| 5–10 | 4 |
| 11–15 | 2 |
| > 15 | 4 |
| **Professional profile** | |
| Nurse | 17 |
| Physician | 2 |
| Social worker | 1 |
| Occupational therapist | 1 |
| Physiotherapist | 1 |
| **Setting of care** | |
| Critical care/Emergency | 6 |
| Nursing home | 5 |
| Medicine | 4 |
| Surgery | 3 |
| Primary care | 3 |
| Hospice | 1 |
| **Work relationship** | |
| Full-time | 19 |
| Part-time | 3 |
| **Education/training courses in clinical communication or counselling** | 3 |

provides the opportunity to discuss own perspective in a collaborative, participatory setting and could be helpful in obtaining viewpoints that are representative of the majority [46]. To develop the educational booklet, the academics strictly worked with stakeholders from other sectors, including professionals with varying scope of practice, representatives of volunteering associations, and associations of patients and family carers with life-limiting illnesses. The iterative, user-centric approach with several rounds of revisions promoted the creation of a person-centered resource tailored on day-to-day practice challenges that is responsive to real life scenarios [35]. For example, in Carlo's case to promote reflection on care choices, the original, professionals-centric sentence *"we have given you a lot of information"* was changed into the person-centric statement *"you have received a lot of information"*. Moreover, the user-centric approach allowed to improve realism by replacing didactic sentences with real-life expressions. Although several communication protocols suggest closing the encounter by making patients/

**Table 3. Acceptability, usefulness, feasibility to use and perceived acquired knowledge.**

| Evaluations of the booklet | N (%) |
|---|---|
|  | N = 22 |
| **Acceptability[a], mean (SD)** | 38 (7) |
| $\geq 30$ | 18 (81.8) |
| $\geq 40$ | 12 (54.5) |
| **Usefulness[b], mean (SD)** | 7.6 (1.8) |
| 6–7 | 6 (27.3) |
| $\geq 8$ | 13 (59.1) |
| **Feasibility to use[b], mean (SD)** | 6.9 (1.6) |
| 6–7 | 9 (40.9) |
| $\geq 8$ | 8 (36.4) |
| **Perceived acquired knowledge[c]** |  |
| Not at all | 1 (4.5) |
| Little | 3 (13.6) |
| Moderately | 12 (54.5) |
| Very | 5 (22.7) |
| Extremely | 1 (4.5) |

[a] Range 10 to 50. Themes addressed: usability, understandability, readability, guidance in providing information to patients and families, communication with patients/families, relationships with patients/families, and intention to use in practice. A score of 30 and higher was deemed "acceptable" and a score of 40 and higher "highly acceptable" (mean item scores 3 and 4, respectively).

[b] Scale from 1 (not at all useful/feasible to use) to 10 (extremely useful/feasible to use). A score between 6 and 7 indicated moderate usefulness/feasibility, while a score of 8 or above indicated a high degree of usefulness/feasibility.

[c] 5-point Likert scale from 1 (not at all) to 5 (extremely).

family carers summarize the main key points discussed, all participants in our focus groups agreed that this is usually performed by the professionals to avoid further pressure. Therefore, the original sentence "*could you try repeating [..]*" was changed into "*we would like to summarize [. . .]*".

As suggested by previous literature on participatory research [47, 48], the co-creation process ensured that the booklet was relevant to the users' priorities and needs and resulted in all professionals recommending the resource to colleagues for its practicality and support in real-case scenarios. The involvement of meaningful stakeholders in quality improvement initiatives is recommended to develop innovative clinical resources aimed at improving practice [49, 50]. An active engagement of potential users in the design of the resources can affect their behavior change more positively, favour the integration of resources in daily practice and is finally more likely to improve health-related outcomes and public health [51].

The I-CVI always above 0.9 and the S-CVI/Ave of 0.97 indicate the high degree of content information validity. The target of a S-CVI/UA above 0.80 was not reached [38]; this was somewhat expected due to the large number of external validators. However, the involvement of 15 external validators despite the fact that more than ten are probably unnecessary [52] was justified by the need to include professionals with different scope of practice who work in a wide array of clinical settings. This approach wants to sustain the core message that serious illness conversations are ubiquitous across several care settings and require a multidisciplinary team [53–55]. Additionally, only two experts deemed some sections not relevant (one 6/17 and one 2/17) (S5 Table).

Table 4. Acceptability of the booklet: Scoring at the item level by the 22 evaluating professionals.

| Items | Score Mean (SD)* | Number indicating 4 or 5 on five-point agreement scale* N (%) |
|---|---|---|
| 1. The booklet is easy to use | 3.7 (1.0) | 11 (50) |
| 2. The booklet is easy to understand | 3.9 (0.8) | 15 (68.2) |
| 3. The booklet is easy to read | 4.0 (0.9) | 18 (81.8) |
| 4. The booklet provides guidance that aligns with the way I think things should be done | 3.8 (1.0) | 15 (68.2) |
| 5. The booklet is a reliable strategy to enhance communication with patients and their families | 3.9 (0.9) | 15 (68.2) |
| 6. The booklet complement my usual approach to communicate with patients and their families | 3.7 (0.9) | 14 (63.6) |
| 7. The booklet will enhance communication between professionals and patients/families | 3.5 (0.9) | 11 (50) |
| 8. The booklet will improve the relationship between professionals and patients/families | 3.5 (0.9) | 11 (50) |
| 9. There is a high probability that using this booklet may cause/result in more benefit than harm | 4 (0.8) | 19 (86.4) |
| 10. I will continue to use this booklet even after the project is completed | 3.6 (0.7) | 13 (59.1) |

*Each item scores from 1 (strongly disagree) to 5 (strongly agree)

The booklet performed well in the product testing of readability and design that employed internationally accepted methods -Gulpease index and BALD criteria, respectively-. In general, an increase in readability leads to a shorter reading time, better memory of the content, and greater text comprehension [56]. Our booklet showed good readability (Gulpease index of 60), was perceived easy to read by over 80% (18/22) of the professionals and 15 (68%) agreed on high understandability. Gray and Leary identified four basic elements that influence the ease of reading: content, style, structure, and design [57]. Design should not be overlooked since sound design principles can help users to understand and use information [58, 59]. Consultation workshops arranged for the development of a resource to support person-centered conversations strongly recommend avoiding an arbitrary design that would reduce the usefulness and uptake among professionals [60]. Multiple adjustments were made to the design and layout of the booklet and a design agency was engaged to improve the production of quick-reading and visual-friendly solutions.

An ever-present barrier to serious illness conversations is the lack of time [34, 54, 61, 62] and professionals need to quickly identify information that answers specific communication challenges. A resource designed to support clinical practice must address this obstacle. In our booklet, each communication scenario is accompanied by practical hints, an infographic, and an example of good interaction. This reflects a time-based approach on conversations templates of 1, 2, and 5 minutes, respectively, according to the amount of time professionals have.

Most professionals perceived the booklet as useful (19/22) and gain in knowledge on how engaging in serious illness conversations (18/22), despite the generally long working experience. This suggests that though experience may be a valid aid in sustaining challenging conversations, it is not enough. An international study involving professionals who work in long-term settings across six European countries found that Italy scored the worst in self-efficacy regarding end-of-life communication [63]. These conversations are emotional and professionals need training and resources to acquire confidence; our booklet is the first resource in Italian language that offers professionals support.

The booklet strives for behaviour change and supports professionals overcoming individual-related barriers and timely initiating these conversations. Our resource is well-placed to promote changes in professional practice as suggested by addressing all the 12 domains identified to explain behavior change [27]. The booklet enhances knowledge to engage successfully in serious illness conversations, provides guidance for real-life difficult communication scenarios while ensuring readability and appealing design the positively influence the memory of the content, is consistent with professional identity and standard, and influences beliefs about capabilities and outcomes with most professionals reporting they would continue to use the resource and perceiving benefits in communication and relationships with families. Additionally, the booklet may stimulate reflection on professional competence and learning needs, enhance sensitivity on the topic of quality and timely communication, which triggers change in behavior, evoke emotions and behaviors related to previous communication experience while providing functional tools for their management, and propose desirable behaviors or instigate a process of social influence.

The booklet supports professionals to sustain serious illness conversations that mirror their most frequent challenges in daily practice and for each scenario offers 1-minute, 2-minute and 5-minute solutions. Communication scenarios are relevant to the professionals' learning needs and priorities to prompt efforts in memorizing information and applying learning that is perceived relevant. Moreover, the opportunity to choose the template that best fits own routine increases the use of the booklet. The easily readable information and the user-friendly design promote the understandability of the booklet content. This helps professionals become more knowledgeable about how engaging in serious illness conversations and supports behavior change to improve practice and patient care [19]. The resource stimulates reflection on professional competence, learning needs, and motivation to enhance skills with reference to the professional standard. The high content information validity while employing communication scenarios set in multiple care settings (i.e., nursing home, oncology, and intensive care unit), as well as the involvement of professionals with different scope of practice and who work in different contexts across the development, validation, and evaluation of the booklet, suggests that the resource may be potentially useful irrespective of the professional profile and the setting of care. Finally, the collaboration between researchers, professionals, and lay stakeholders and the testing of the booklet using validated tools ensured the development of a resource of increased quality and utility.

The booklet presents a limited number of advanced chronic or terminal illnesses, but addresses the most frequent communication scenarios across different clinical settings and offers practical tips and examples of communication protocols that can be tailored to different chronic conditions. Moreover, issues related to communication with adults about an infant or child's illness are not addressed. Parents are often scared, feel guilty, and may have difficulties retaining and processing information. When communicating to parents, professionals have to listen authentically, avoid judgement and act empathically to capture their history, values, perspectives and peculiarities that make one family different from another. Parents need to be supported and accompanied in the experience of their child's illness, beyond receiving information that should be tailored with regard to content and modalities. Therefore, a specific resource to support communication in this setting is warranted. Secondly, the involvement of the design agency occurred late in the development process. Its integration from the outset of the project could have eased addressing the feedbacks and translating them into solutions. However, the design agency had received information on the objectives of the booklet and the main critiques emerged during its development, and the research team provided ongoing supervision. Thirdly, the evaluation of the booklet was mainly quantitative in nature, with professionals reporting ratings on acceptability, usefulness, feasibility of use, and acquire

knowledge; despite professionals had the opportunity to report open comments and suggest changes, qualitative work could have contributed more deeply to identify nuances that benefit from improvements.

## Conclusions

The booklet developed helps address the lack of support tools on serious illness conversations for Italian professionals and could potentially improve the quality of serious illness conversations.

The co-creation process yielded an acceptable resource with good readability and excellent design, high content validity, and a high degree of perceived usefulness and acquired knowledge. Also, the time-based solutions favor the use of the resource in daily practice. All these features make the booklet ideally positioned to encourage changes in professional practice and a valid resource to enhance shared decision-making and care planning. In addition, the booklet has good potential for a wider distribution, as it is available on the website of a high-profile Italian society of palliative care and can be downloaded for free. Finally, translation to English and social media campaigns beyond congress presentations are planned to promote dissemination.

## Supporting information

**S1 Table. Standards for Quality Improvement Reporting Excellence for Education (SQUIRE-EDU).**
(PDF)

**S2 Table. Review of the principles or content of the main communication protocols and methods.**
(PDF)

**S3 Table. Characteristics of the participants to the focus groups (n = 15).**
(PDF)

**S4 Table. Characteristics of the external review board (n = 15).**
(PDF)

**S5 Table. Clarity, coherence, relevance, and readability for each sub-section of the booklet.**
(PDF)

**S6 Table. Layout and design characteristics of the booklet based on the Baker Able Leaflet Design (BALD) criteria.**
(PDF)

**S1 Graphical abstract.**
(TIF)

## Acknowledgments

We would like to express our gratitude to all professionals and associations engaged in the development, validation, and evaluation of the booklet for their cooperation. We thank Ylenia Arnone and Alessio Turrini for their contribution in developing the booklet content, Alexandra Antal and Valentino Bissacca for their help in validating the resource, and Luisa Goglio for revising its design. Finally, we thank Dr. Nicola Cornally for her methodological guidance across the development and validation processes.

## Author Contributions

**Conceptualization:** Silvia Gonella, Paola Di Giulio, Federica Riva-Rovedda.

**Data curation:** Silvia Gonella.

**Formal analysis:** Silvia Gonella.

**Funding acquisition:** Valerio Dimonte.

**Investigation:** Silvia Gonella, Federica Riva-Rovedda.

**Methodology:** Silvia Gonella, Paola Di Giulio, Federica Riva-Rovedda, Valerio Dimonte.

**Project administration:** Silvia Gonella.

**Resources:** Luigi Stella, Maria Marcella Rivolta, Eugenia Malinverni, Mario Paleologo, Giancarlo Di Vella.

**Supervision:** Valerio Dimonte.

**Validation:** Silvia Gonella.

**Visualization:** Silvia Gonella.

**Writing – original draft:** Silvia Gonella.

**Writing – review & editing:** Silvia Gonella, Paola Di Giulio, Federica Riva-Rovedda, Luigi Stella, Maria Marcella Rivolta, Eugenia Malinverni, Mario Paleologo, Giancarlo Di Vella, Valerio Dimonte.

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
