## [Decision Letter · Decision Letter 0]

17 Mar 2024

PONE-D-24-04935Sustaining health and social care professionals in serious illness conversations: Development, validation, and preliminary evaluation of an educational bookletPLOS ONE

Dear Dr. Gonella,

Thank you for submitting your manuscript to PLOS ONE. After careful consideration, we feel that it has merit but does not fully meet PLOS ONE’s publication criteria as it currently stands. Therefore, we invite you to submit a revised version of the manuscript that addresses the points raised during the review process.

We look forward to receiving your revised manuscript.

Kind regards,

Elvan Wiyarta, M.D.

Academic Editor

PLOS ONE

- https://doi.org/10.3390/healthcare12030398

In your revision ensure you cite all your sources (including your own works), and quote or rephrase any duplicated text outside the methods section. Further consideration is dependent on these concerns being addressed.

“This work was supported by Fondazione Assistenza e Ricerca Oncologica (F.A.R.O.) Onlus. The funding did not influence the study process either the study findings.”

6. Please include a separate caption for each figure in your manuscript.

7. Please include a copy of Tables 1, 2, 3 and 4 which you refer to in your text on page 10 and 11.

Additional Editor Comments:

Please revise according to the reviewer's input

Reviewers' comments:

Reviewer's Responses to Questions

**Comments to the Author**

1. Is the manuscript technically sound, and do the data support the conclusions?

Reviewer #1: Partly

Reviewer #2: Yes

2. Has the statistical analysis been performed appropriately and rigorously? 

Reviewer #1: Yes

Reviewer #2: Yes

3. Have the authors made all data underlying the findings in their manuscript fully available?

Reviewer #1: No

Reviewer #2: Yes

4. Is the manuscript presented in an intelligible fashion and written in standard English?

Reviewer #1: Yes

Reviewer #2: Yes

5. Review Comments to the Author

Reviewer #1: Review: Sustaining health and social care professionals in serious illness conversations: Development, validation, and preliminary evaluation of an educational booklet

Abstract

Clarity required about what constitutes a serious illness conversation (i.e. what illness, at what stage of illness trajectory) and whether the patient is an adult or a child. It is not clear from the abstract who (assume healthcare professionals) is aiming to communicate to who (the patient, patient’s family etc)

Introduction

Please make it clearer much earlier who is communicating to who (as per comments on abstract)

Communication skills training may not be a part of education for healthcare professionals in Italy, but is certainly a growing part of formal professional training for healthcare professionals in other countries. Please acknowledge these differences; the introduction could be strengthened by considering how other healthcare systems attempt to address challenges of communication.

The introduction helpfully identifies literature regarding the benefits of written communication guides; these are based on a wide range of different clinical contexts and the section could be strengthened by examination of where these have been used (eg with frail elderly vs acute clinical contexts) and the implications of these different environments for the nature and type of communication.

This section needs a much stronger rationale as to the limitations of the guides which exist already, and why something new is required.

The paper requires a rationale about why it was decided to use the format of a written communication booklet. Were other modalities considered eg video, audio, online learning etc.

Method

Please elaborate on the rationale for the clinical cases used and the communication challenges selected to be illustrated in these scenarios.

It would enhance the paper to include a brief review of the communication protocols referenced as examples of ‘good communication scripts’, to include their main principles or content.

Please clarify why representatives from patient associations, carers and volunteers were only drawn from the dementia field, how these representatives were identified and recruited.

Please add additional methodological information about the feedback from the focus group participants and how this was integrated into the second iteration of the booklet. It is essential that the methodology for analysing the results of the discussion is clearly described, how potential conflicts of opinion were resolved etc. Please add the ‘pre-defined set of questions’ as a table or to the body of the manuscript, with a rationale for how these were selected. If the focus groups were not recorded, transcribed and analysed using an established methodology, please state and explain why not.

Validation: please add further detail about the validation process. In particular, further information is required regarding the number of sections which did not achieve an I-CVI of >0.70 and what decisions were made regarding improving these sections or discarding them. Were there differences between the professional raters and those from patient organisations?

Further information is required about the sections which were adjusted, detailing what needed to be addressed to improve relevance, clarity or realism and how this was addressed.

The statistical analyses used a methodology based on decision aid development and evaluation. Please explore the similarities and differences between a decision aid and a communication booklet.

I suggest that presenting the method and then results for each of the phases of development would enhance the readability of the manuscript.

Results

The amendments to the initial draft of the guide is well documented in Table 1. However, this does not include adequate detail about why these adjustments were made (see comment re methodology used to identify and make decisions about what to amend).

Discussion

The authors highlight the importance of developing a resource to support person-centred conversations; please add in reflection about the decision not to use A Person-Centred Approach methodology. Elements of this methodology have been used by the team, but there is no mention of this within the paper.

A substantial section of the discussion is framed around the 12 domains of behaviour change. This is a useful framework, but it would be more logical and helpful to include this in the introduction; currently it feels like a ‘post hoc’ analysis rather than being core to the development of the booklet.

The decision to involve a design agency is not included in the method an needs earlier integration into the paper, with a clear rationale about the problem being addressed.

The discussion briefly mentions ‘the maternal-infant context’. I assume this refers to talking to adults when the patient is an infant or child? Please clarify the language around this. The issues relating to communication about a child’s illness are related but also include separate, additional challenges. This needs much more careful consideration than a single line in the discussion.

Please expand the section discussing the limitations of the booklet and its development. For example (but not exclusively), would any qualitative work have contributed to the evaluation of the booklet? What are the pros and cons of a physical booklet for wider distribution? How are the authors planning to disseminate the booklet? What impact do they anticipate it would have on patient care and how could this be evaluated? Limitations of the patient scenarios given the broader context of different types of chronic illness?

The booklet is aimed at discussions with patients, but particularly in the context of frail elderly or patients who are at end of life, what are the implications about communication with other members of the family or carers?

Reviewer #2: The introduction was comprehensive and read well.

Line 113 and contributes to know the patients’ valued priorities and align care with their preferences.

Methodology included creation of the resource, validation and evaluation.

Line 194

Fourthly, the booklet was further adjusted according to the emerging comments

Could the authors please provide the pre-set questions used in the focus group?

Results were provided with appropriate statistics.

A link to the booklet itself is provided. I would be interested to read an English version, if available.

This included discussion of the co-creation process.

The option of communication interventions appropriate for 1, 2 and 5 minutes was an excellent concept.

The paragraph on page 14 is long and could be divided into 2.

Suggest add 2 - 3 sentences to the conclusion.

6. PLOS authors have the option to publish the peer review history of their article (what does this mean?). If published, this will include your full peer review and any attached files.

Reviewer #1: No

Reviewer #2: **Yes: **Dr Anthony Herbert

---

## [Author Response · Author response to Decision Letter 0]

5 Apr 2024

Please, see the file "Response to reviewers".

---

## [Decision Letter · Decision Letter 1]

8 May 2024

Supporting health and social care professionals in serious illness conversations: Development, validation, and preliminary evaluation of an educational booklet

PONE-D-24-04935R1

Dear Dr. Gonella,

We’re pleased to inform you that your manuscript has been judged scientifically suitable for publication and will be formally accepted for publication once it meets all outstanding technical requirements.

Kind regards,

Elvan Wiyarta, M.D.

Academic Editor

PLOS ONE

Additional Editor Comments (optional):

Reviewers' comments:

Reviewer's Responses to Questions

**Comments to the Author**

1. If the authors have adequately addressed your comments raised in a previous round of review and you feel that this manuscript is now acceptable for publication, you may indicate that here to bypass the “Comments to the Author” section, enter your conflict of interest statement in the “Confidential to Editor” section, and submit your "Accept" recommendation.

Reviewer #2: All comments have been addressed

2. Is the manuscript technically sound, and do the data support the conclusions?

Reviewer #2: Yes

3. Has the statistical analysis been performed appropriately and rigorously? 

Reviewer #2: Yes

4. Have the authors made all data underlying the findings in their manuscript fully available?

Reviewer #2: Yes

5. Is the manuscript presented in an intelligible fashion and written in standard English?

Reviewer #2: Yes

6. Review Comments to the Author

Reviewer #2: I have reviewed this paper previously.

The authors have addressed all of my suggested comments.

This includes an expanded conclusion.

7. PLOS authors have the option to publish the peer review history of their article (what does this mean?). If published, this will include your full peer review and any attached files.

Reviewer #2: No

---

## [Editor Report · Acceptance letter]

22 May 2024

PONE-D-24-04935R1 

PLOS ONE

Dear Dr. Gonella, 

I'm pleased to inform you that your manuscript has been deemed suitable for publication in PLOS ONE. Congratulations! Your manuscript is now being handed over to our production team.

Kind regards, 

on behalf of

Mr. Elvan Wiyarta 

Academic Editor

PLOS ONE